# Joint Value as a Measure of Sea Trade Port Stakeholder Effect

**Iryna Nyenno** [1,*] **, Natalia Rekova** [2] **and Svetlana Minakova** [3]

1    Department of Management and Innovations, Odessa National University after I.I. Mechnikov, 65082 Odessa, Ukraine
2    Department of Enterprise Economics, Donbass State Engineering Academy, 84313 Kramators'k, Ukraine; natarekova@gmail.com
3    Chernovetskii Department of Economics and Management, National Technical University "Kharkiv Polytechnic Institute", 65009 Chernivtsi, Ukraine; smmnkv@gmail.com
*    Correspondence: inyenno@onu.edu.ua; Tel.: +380-50-585-6574

**Abstract:** This article is devoted to an efficiency measurement of the maritime industry presented through the joint value of industry stakeholders. A list of factors contributing to the efficiency of the state maritime policy and factors in the development of the maritime industry were defined and separated into four groups: group 1 (infrastructural factors): Renewal of port infrastructure; coastal infrastructure of sea stations; ecological and physical safety; and convenience in reaching the port of departure of a cruise ship; group 2 (management factors): The effectiveness of management mechanisms; the level of automatization and effective communications technologies; the coordination of various types of transport; and the efficiency of port services; group 3 (marketing factors): Tariff policies (tariff amounts, number of port fees, flexibility of the price policy); and competition in the ports; group 4 (service factors): Attractiveness of logistics conditions; the development of international tourism; the development of sea leisure; the development of merchant shipping, shipbuilding, ship repair, and instrument making in the port; and the simplification of port entry procedures. The joint value was considered to be a category at both a macroeconomic and microeconomic level, and it was combined with a multivariate regression model performed on the basis of the statistical analysis and data processing system Statistica 8.0. The complex combination of the results of the multifactorial linear model of the joint value created in the maritime industry led to the conclusion that the best alternative to the development of the port industry in Ukraine is the scenario of state modernization and corporatization in the port business model.

**Keywords:** maritime industry; port business model; joint value; stakeholder effect

## 1. Introduction

Thanks to favorable market conditions, the maritime industry development forecast according to the World Maritime Review 2018 (World Maritime Review 2018) is enthusiastic and optimistic. World trade is emerging, while domestic economies are more and more demanding of international products and export impacts. This industry is so very attractive for different role players, such as transport and logistics, crewing companies, marine ports, exporters, and importers, that the issue of its measurement is debatable and has a lot of distinctive ways of implementation. In this article, a new approach to evaluating the effect created by the maritime sector is described. Namely, the relevant effect of different stakeholders is united to check the synergy.

The joint value created by the seaport takes into account the fact that as a result of the activity of each entity, there should be a maximization of utility in all stakeholders effectively performing

specified roles. These stakeholders are the state, investors, owners, contractors of ports, the population of the port region, regular workers, and invited specialists. At the same time, sustainable development in the maritime industry is achieved by the integration of global value chains through international integration, mergers and takeovers of ports, and the creation of cross-border companies in particular.

Maximization of the usefulness of stakeholders is possible through optimization of the impact of positive and negative externalities in the maritime industry. Among the positive externalities are internationalization, the development of sea tourism and leisure, renovation of coastal infrastructure, simplification of customs clearance, the removal of administrative obstacles, the development of merchant shipping, shipbuilding, ship repair, instrument making, construction of navigable canals, bridges, staff training, employment, and wages. Meanwhile, the negative externalities are competition in ports; increased probabilities of violations of ecological and physical security in the port; the safety of transported cargo, passengers, and luggage; the complexity of coordination of work; and the interactions between different types of transport, cargo owners, forwarders, and stevedores in the port.

To prevent the formation of negative externalities, the common value forms conditions for achieving high ecological status and protection of the resources upon which the economic maritime profit-making activity is based. Thus, within the framework of the concept of sustainable development of ports, the necessary elements are an economic component (regional and macroeconomic development), a social component (raising the living standards of the seaport industry staff), and an ecological component (meeting the requirements of environmental standards). The shared value as a value added process for sustainable development motivates the implementation of an integrated approach for all authorities and management at the sectoral and administrative levels in the maritime industry, which allows for systematic efforts toward synergy and for identifying differences and inefficiencies. Efficiency is a concept that aims for the optimal use of resources. Its relation to the economy is obvious: Best-quality services and products provided with minimal resource use. Stakeholders, for whom joint value is created, influence organization significantly. The joint value rising influences the level of sustainable development.

Efficiency is improved when more outputs of a given quality are produced with the same or fewer resource inputs, or when the same amount of output is produced with fewer resources. For an in-depth explanation of the relationship between the efficiency concept and the joint value effect, it is worth analyzing the understanding of effect and efficiency. The joint value effect in the maritime industry is concerned with the extent of the achievement of stakeholder goals. Meanwhile, in the sense of the efficiency concept, the joint value of the maritime industry is considered to be a united dimension that demonstrates synergetic initiatives and strong partnership energy for the industry's potential development. Efficiency is achieved through a collaborative approach measured by the joint value effect.

## 2. Literature Review

Traditionally, port efficiency has been assessed by methods measuring cargo handling productivity. One of them is a factor based on productivity, while the other one is based on comparing tangible conditions with optimum throughputs over a certain period of time. Moreover, there are also methods originating from the estimation of a port cost function and the computing of total factor productivity. In addition, multiple regression analysis based on port performance and efficiency estimation models have been applied in the literature. However, methods based on calculations of relative efficiency with respect to productive activities have been growing recently (Gokcek and Şenol 2018).

Kelly and Alam (2008) have concluded that managers, investors, and all concerned must accept that in the contemporary world, the wealth of a corporation is not merely the property it owns, the financial resources it accumulates, or even the intellectual property it develops.

MacManus, J., has separated the intrinsic value of stakeholders based on behavioristic characteristics: Trust, motivation, success, relationship, and influence (MacManus 2002). Turki et al. (2012) have emphasized that there is a stakeholder value network. Ping Wang has mentioned

that recently the maritime industry has been actively developed through the study of managerial disciplines, such as business logistics and strategic management. The maritime economy needs more and more managerial tools to be adequately measured. Business model research takes place with the usage of descriptive and constructive definitions. The constructive ones are concentrated on system building (Osterwalder and Pigneur 2010; Vashakmadze 2012), while descriptive ones set up the characteristics of this category. A range of authors have considered the external impact of the economic environment on modern business models. They have researched their characteristics. The attention of such scientists as Chesbrough (2007), Schweizer (2005), Debelak (2006), and Slywotskiy and Morrison (1997) has been directed to the classification approaches of business models. One still-unsolved task is the introduction of a practical approach to business model analysis. Bereznoy (2014) has emphasized the interdisciplinary nature of the term "business model". This nature explains the incomplete processing of this term in the literature on economic theory, a theory of strategy and organization where the business model is mentioned without giving a precise definition. Summarizing the approaches to "business model" definitions, Soolyatte (2010) noted that major differences in the interpretation of the term "business model" occur among technology-oriented people and business-oriented people. (Slywotskiy and Morrison 1997), in his research "Migration value. What will happen with your business tomorrow?", explored how value migrates due to changes in business models. The purpose of the sea trade port business model is added value creation (Nyenno et al. 2017). Meanwhile, the business model itself ignores the environmental factors of stakeholder influence on value. Thus, we offer here a description of an updated measure of the maritime industry effect called "stakeholders' joint value". The research hypothesis was that joint value created in the maritime industry may evaluate the stakeholder effect related to industry alternative development scenarios. Carlon and Downs (Carlon and Downs 2014) have suggested a stakeholder valuing process by valuing customers with tax expenses (including profit tax). They accounted for primary stakeholders, employees, and customers as intangible assets that are capitalized at historical values and amortized over expected lives (valued by staff costs, social fare, and profit). In other words, stakeholders are reported as assets on the firm's balance sheet, and amortization of the assets is reported as an expense on the income statement. In a continuation of this approach to profit tax, net profit, staff costs, social fare, and depreciation were hypothetically considered to measure the joint value of maritime industry stakeholders. According to the research hypothesis, the elements listed below are relevant to the effect obtained by the following maritime industry stakeholders: The state, the population of the port region (profit tax), investors and owners (net profit, depreciation), contractors of ports, regular workers, and invited specialists (staff costs, social fare).

## 3. Methodology

The overall methodology used in the research to generate results had the following six steps:

1. Justifying the element of the joint value;
2. Finding the formula for the joint value of the multifactorial linear model in the maritime industry;
3. Verifying the formula of the joint value with extrapolation;
4. Verifying the formula of the joint value by using a Fourier series;
5. Using the method of hierarchy analysis for an indication of the influence of objective and subjective factors on joint value;
6. Evaluating the result of the joint value in the maritime industry related to the following alternatives: A scenario of state modernization and corporatization; a scenario for managing the seaport as a concession (public–private partnership); a scenario for changing the owner of the port subject to attracting private capital; a scenario of inertial development of the port provided there is stagnation in the macroeconomic environment.

Through the value of the "the joint value" created in the maritime industry and through constructing a multifactorial model, the following factors significant to our research were separated, namely:

$x_{1t}$: Profit tax;

$x_{2t}$: Net profit;

$x_{3t}$: Staff costs;

$x_{4t}$: Social fare; and

$x_{5t}$: Depreciation.

Thus, the research model was based on a validity check of this hypothesis first by verification with the extrapolation and next by using a Fourier series in the case of low dependencies. For further investigation of the influence of objective and subjective indicators on the indicator of the joint value of the seaport services, we made our choice in the use of the method of expert evaluation, namely, the method of analysis of hierarchies.

A general view of the multifactorial model of joint value created in the maritime industry can be represented as follows:

$$y_t = b_0 + b_1 x_{1t} + b_2 x_{2t} + \ldots + b_m x_{mt} + \varepsilon_t, t = 1, \ldots, n, \tag{1}$$

where $y_t$ is the joint value created in the port maritime industry (result indicator); $b_0$ is an unknown parameter (coefficient) that reflects the influence of exogenous macrofactors; $b_1, b_2, \ldots, b_m$ are unknown parameters (coefficients) reflecting the influence of endogenous microenvironment factors; $x_{1t}, x_{2t}, x_{3t}, \ldots, x_{mt}$ are independent variables (factors) in thousands of UAH; $\varepsilon_t$ is a random component; and $n$ is the number of observations.

A statistical analysis of the source data and a construction of a multifactor model of joint value created in the maritime industry was carried out by using a software package for statistical analysis, Statistics 8.0. We present the source data for a multifactor model of joint value created in the maritime port sector (Table 1).

Given the independent variables of the model (income tax on general activities, net profit, payroll, social contributions, depreciation) and the result indicator (joint value) were measured by thousands of hryvnias, there was no need to perform data normalization. The linear multifactor regression involved linear relationships between variables and the subordination of model balances to the normal distribution law.

The linearity of the relations between variables of the multifactor linear model of joint value created in the maritime industry was proven and is presented in the form of a matrix of scattering diagrams in Figure 1.

A verification of the subordination of the residues of the multifactorial linear model of joint value created in the maritime industry to the normal distribution law is presented in Figure 2.

The results obtained graphically in Figures 2 and 3 show the correct choice of mathematical dependence for a multifactor model of joint value created in the maritime industry, namely a linear one.

The calculated values of the free member and the regression coefficients of the multivariate linear model of joint value created in the maritime industry sector were done in Statistica 8.0. The package was obtained and is presented in Table 2.

**Table 1.** Output data for a multifactor model of joint value created in the maritime industry, in thousands of UAH.

| No. | Profit Tax | Net Profit | Staff Costs | Social Fare | Depreciation | Joint Value of the Maritime Industry |
|---|---|---|---|---|---|---|
| | $X_1$ | $X_2$ | $X_3$ | $X_4$ | $X_5$ | y |
| 1 | 2 | 3 | 4 | 5 | 6 | 7 |
| 1 | 4973.11 | 16,417.76 | 35,095.67 | 9435.48 | 2268.03 | 56,486.54 |
| 2 | 7607.69 | 26,722.53 | 57,470.77 | 15,788.36 | 3469.56 | 91,800.98 |
| 3 | 9976.79 | 35,631.41 | 76,009.34 | 21,051.96 | 4778.73 | 121,617.54 |
| 4 | 12,537.84 | 44,778.00 | 93,179.00 | 26,456.00 | 5718.00 | 150,494.84 |
| 5 | 645.67 | 16,632.98 | 37,262.80 | 12,722.39 | 2928.05 | 54,541.45 |
| 6 | 931.67 | 28,433.79 | 56,550.15 | 18,983.66 | 4700.70 | 85,915.61 |
| 7 | 1263.45 | 37,459.45 | 78,615.61 | 24,895.35 | 5947.93 | 117,338.51 |
| 8 | 1587.78 | 45,365.00 | 96,374.00 | 31,286.00 | 7382.00 | 143,326.78 |
| 9 | 2162.52 | 5415.82 | 10,835.63 | 3140.71 | 756.01 | 18,413.97 |
| 10 | 3253.63 | 8868.65 | 16,029.60 | 5112.00 | 1156.52 | 28,151.89 |
| 11 | 4392.87 | 11,441.87 | 23,103.78 | 6816.26 | 1592.91 | 38,938.51 |
| 12 | 5452.00 | 14,379.00 | 27,318.00 | 8566.00 | 1906.00 | 47,149.00 |
| 13 | 3940.00 | 11,908.00 | 24,268.00 | 8622.00 | 2282.00 | 40,116.00 |
| 14 | 6442.00 | 17,147.00 | 42,790.00 | 15,623.00 | 4551.00 | 66,379.00 |
| 15 | 8817.00 | 21,648.00 | 59,919.00 | 22,215.00 | 6800.00 | 90,384.00 |
| 16 | 11,366.00 | 23,699.00 | 78,169.00 | 29,124.00 | 9110.00 | 113,234.00 |
| 17 | 0.00 | 10,482.00 | 22,028.00 | 8293.00 | 2349.00 | 33,700.00 |
| 18 | 0.00 | −15,065.00 | 40,207.00 | 15,063.00 | 11,445.00 | 25,142.00 |
| 19 | 0.00 | −21,755.00 | 51,983.00 | 19,795.00 | 19,188.00 | 30,228.00 |
| 20 | 1190.00 | −28,568.00 | 67,521.00 | 25,612.00 | 25,769.00 | 40,143.00 |
| 21 | 0.00 | 6552.00 | 17,831.00 | 6562.00 | 5072.00 | 24,383.00 |
| 22 | 0.00 | 29,058.00 | 34,655.00 | 12,780.00 | 10,095.00 | 63,713.00 |
| 23 | 0.00 | 47,289.00 | 52,440.00 | 19,335.00 | 15,241.00 | 99,729.00 |
| 24 | 14,131.00 | 48,191.00 | 81,007.00 | 28,789.00 | 20,511.00 | 143,329.00 |
| 25 | 15,649.27 | 103,532.59 | 33,323.22 | 11,688.17 | 8260.44 | 152,505.07 |
| 26 | 26,325.32 | 139,338.02 | 51,938.62 | 17,767.38 | 13,659.09 | 217,601.96 |
| 27 | 33,536.73 | 176,803.70 | 76,147.20 | 27,549.20 | 18,770.32 | 286,487.63 |
| 28 | 42,682.00 | 187,400.00 | 89,952.00 | 31,931.00 | 23,568.00 | 320,034.00 |
| 29 | 12,049.34 | 73,491.01 | 37,910.57 | 7153.98 | 8245.85 | 123,450.92 |
| 30 | 19,411.44 | 100,237.17 | 58,065.26 | 12,369.00 | 14,531.68 | 177,713.87 |
| 31 | 25,456.35 | 128,161.84 | 81,513.05 | 17,945.21 | 22,129.31 | 235,131.24 |
| 32 | 31,991.00 | 133,023.00 | 102,335.00 | 22,636.00 | 27,441.00 | 267,349.00 |
| 33 | 10,570.63 | 53,039.43 | 40,198.30 | 8514.77 | 10,123.03 | 103,808.36 |
| 34 | 14,538.01 | 78,526.71 | 66,229.94 | 13,691.53 | 17,917.33 | 159,294.66 |
| 35 | 19,962.88 | 93,746.51 | 93,658.61 | 19,506.97 | 24,988.00 | 207,368.00 |
| 36 | 24,776.00 | 101,517.00 | 114,703.00 | 24,606.00 | 32,603.00 | 240,996.00 |

The following mathematical formula for the multifactorial linear model of joint value created in the maritime industry was received:

$$y_t = 387.676 + 1.0052x_{1t} + 1.0021x_{2t} + 0.9295x_{3t} + 0.1753x_{4t} - -0.0056x_{5t} + \varepsilon_t. \tag{2}$$

The next step in the process of extrapolation of the result indicator of the multifactor linear model of joint value created in the maritime industry was analytical alignment and the choice of an adequate model for the subsequent extrapolation process of the dynamic series of its independent variables. The verification process covered only those types of analytical dependencies that visually repeated the trend of development of the index (independent variable model) over time (visual inspection) and a determination coefficient that exceeded 95% (Table 3).

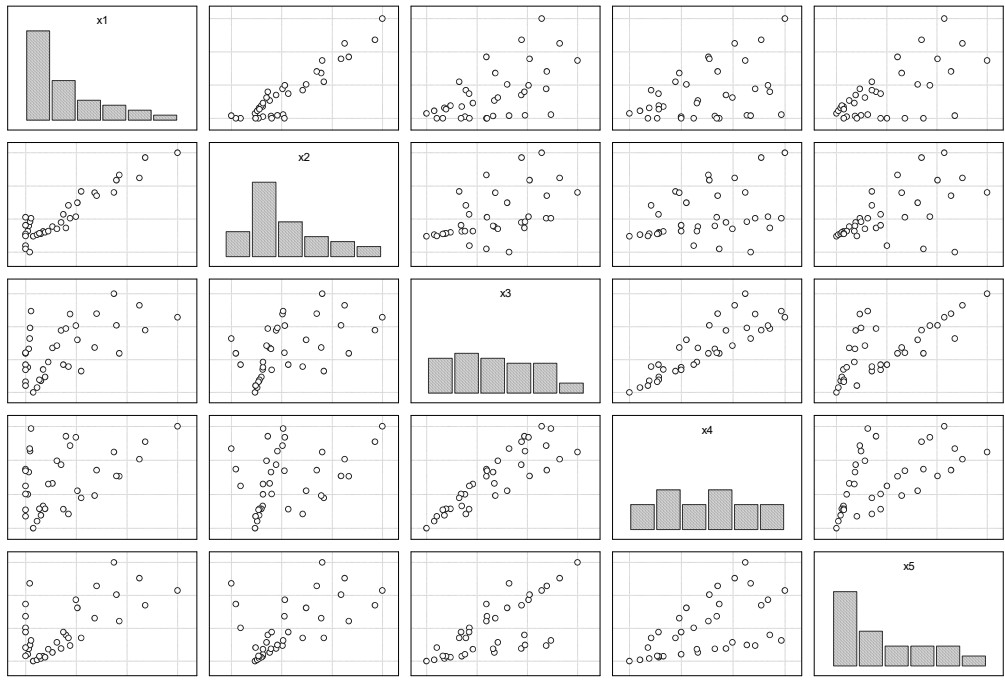

**Figure 1.** A matrix of scattering diagrams of independent variables of a multifactorial linear model of joint value created in the maritime port sector (in the package Statistics 8.0).

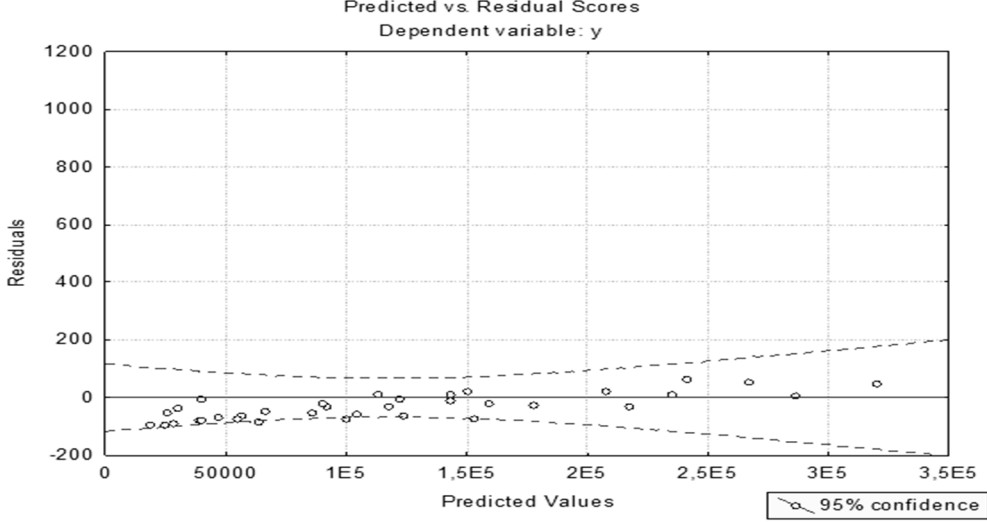

**Figure 2.** The subordination of the residues of the multifactorial linear model of joint value created in the maritime port sector to the normal distribution law (in Statistica 8.0).

**Table 2.** Estimated volume of the free member and regression coefficients of the multifactor linear model of joint value created in the maritime industry.

| Conditional Symbol of the Coefficient | Qualitative Volume of the Coefficient |
|:---:|:---:|
| $b_0$ | 387.6760 |
| $b_1$ | 1.0052 |
| $b_2$ | 1.0021 |
| $b_3$ | 0.9295 |
| $b_4$ | 0.1753 |
| $b_5$ | $-0.0056$ |

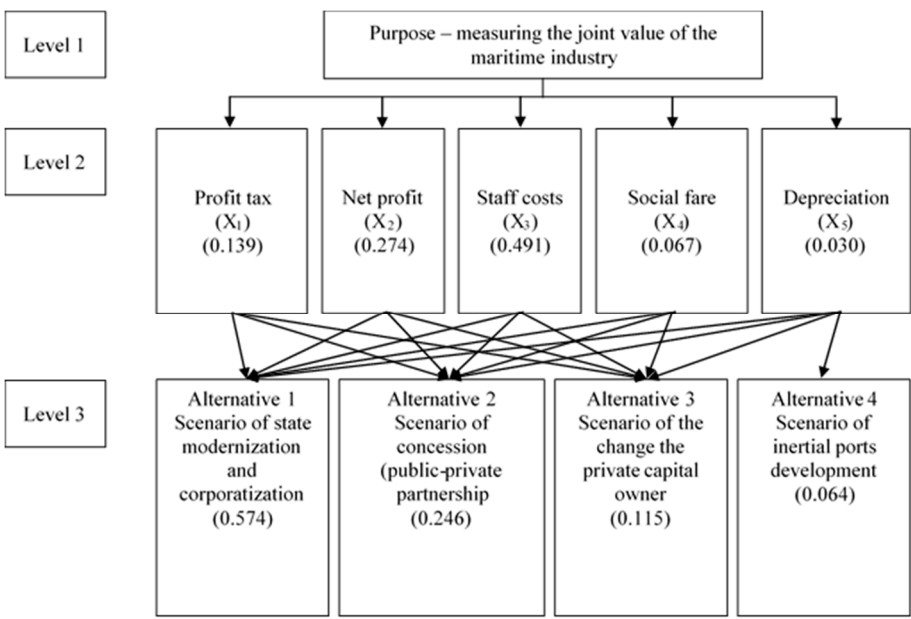

**Figure 3.** Hierarchy model for evaluation of the joint value of the maritime industry with the fixed global priorities of industry development.

**Table 3.** Results of visual and statistical verification of analytical dependencies of independent variables of a multifactor linear model of joint value created in the maritime port sector.

| The Name of the Independent Variable | Type of Analytical Dependence | The Result of the Visual Check (Passed/Not Passed) | Result of Verification of the Value of the Determination Factor (Performed/Not Performed) | |
|---|---|---|---|---|
| | | | Numerical Value of the Determination Factor | Does It Exceed 95% (Yes/No)? |
| 1 | 2 | 3 | 4 | 5 |
| Profit tax | Polynomial 1 degree | Not passed | 0.3670 | No |
| | Polynomial 2 degree | Not passed | 0.4522 | No |
| | Polynomial 3 degree | Not passed | 0.5038 | No |
| | Degree | Impossible to build | - | - |
| | Logarithmic | Not passed | 0.1909 | No |
| | Exponential | Impossible to build | - | - |
| Net profit | Polynomial 1 degree | Not passed | 0.3778 | No |
| | Polynomial 2 degree | Not passed | 0.4601 | No |
| | Polynomial 3 degree | Not passed | 0.5370 | No |
| | Degree | Impossible to build | - | - |
| | Logarithmic | Not passed | 0.2051 | No |
| | Exponential | Impossible to build | - | - |
| Staff costs | Polynomial 1 degree | Not passed | 0.0946 | No |
| | Polynomial 2 degree | Not passed | 0.2756 | No |
| | Polynomial 3 degree | Not passed | 0.2764 | No |
| | Degree | Not passed | 0.0223 | No |
| | Logarithmic | Not passed | 0.0279 | No |
| | Exponential | Not passed | 0.0890 | No |
| Social fare | Polynomial 1 degree | Not passed | 0.0107 | No |
| | Polynomial 2 degree | Not passed | 0.0135 | No |
| | Polynomial 3 degree | Not passed | 0.0210 | No |
| | Degree | Not passed | 0.0091 | No |
| | Logarithmic | Not passed | 0.0078 | No |
| | Exponential | Not passed | 0.0204 | No |
| Depreciation | Polynomial 1 degree | Not passed | 0.5906 | No |
| | Polynomial 2 degree | Not passed | 0.6168 | No |
| | Polynomial 3 degree | Not passed | 0.6191 | No |
| | Degree | Not passed | 0.3985 | No |
| | Logarithmic | Not passed | 0.4100 | No |
| | Exponential | Not passed | 0.5585 | No |

The obtained results indicated the appropriateness of the use of a Fourier series for the process of approximation and extrapolation of the dynamic series of independent variables.

The general mathematical expression of the Fourier series can be written as follows:

$$
\begin{aligned}
&x(t) = \frac{a_0}{2} + \sum_{k=1}^{n} (a_k \cos(kt) + b_k \sin(kt)), \\
&\Delta e \; a_0 = \frac{1}{\pi} \cdot \int_{-\pi}^{\pi} x(t)dt, \; a_k = \frac{1}{\pi} \cdot \int_{-\pi}^{\pi} x(t) \cos ntdt, \\
&b_k = \frac{1}{\pi} \cdot \int_{-\pi}^{\pi} x(t) \sin ntdt, \; u = t + 1 \Rightarrow du = dt, \\
&dv = \cos ntdt \Rightarrow v = \int \cos ntdt.
\end{aligned}
\tag{3}
$$

where $a_0, a_k, b_k$ are coefficients of the Fourier series.

The MathCad package was chosen to determine the coefficients of the Fourier series. The reason for choosing the MathCad package was the universality of its solutions, its high-accuracy error detection, and its visibility and ability to interact with other software products, including products designed to build a risk map and analyze the total risk of a bankrupt merchant marine port using the spectrographic method (which requires collaboration with Microsoft Excel). MathCad has the ability to get a documented metric calculation, that is, to analyze the results in stages. All of the above-mentioned reasons simplify both the collection of data and their subsequent verification.

The process of determining the coefficients of the Fourier series for the indicator "tax on profit from general activity, ths. UAH" in the package MathCad is presented in Figure 3. Similarly, a calculation of the Fourier series coefficients for other independent variables of the multifactor linear model of common value created in the marine port sector was made, and analytical dependencies were obtained.

We performed a verification of the obtained analytical dependencies for the investigated independent variables for adequacy and accuracy of the forecast. The statistical indicators of the adequacy of the analytical dependencies of the independent variables of the multifactor linear model of common value created in the marine port sector are presented in Table 4.

**Table 4.** Statistical indicators of the adequacy of the obtained analytical dependencies (Fourier series) of independent variables of the multifactor linear model of joint value created in the maritime industry.

| No. | The Name of the Statistical Indicator of the Adequacy of the Analytical Dependence | Mathematical Designation of an Independent Variable of a Multifactor Linear Model of Joint Value Created in the Maritime Port Sector | | | | |
| --- | --- | --- | --- | --- | --- | --- |
| | | $X_1$ | $X_2$ | $X_3$ | $X_4$ | $X_5$ |
| | | Numerical Value of the Statistical Indicator of the Adequacy of the Analytical Dependency | | | | |
| 1 | Determination factor, % | 0.999973396 | 0.999995508 | 0.999980087 | 0.999929597 | 0.999953397 |
| 2 | Average relative error of approximation, % | 4.7512 | 2.8243 | 1.1927 | 3.2840 | 4.2215 |
| 3 | Average absolute error, % (*MAPE*) | 17.58 | 10.62 | 10.05 | 12.17 | 15.81 |

The obtained results indicated the high accuracy of the obtained analytical dependencies of the independent variables of the multifactor linear model of the joint value created in the marine port industry, since the value of the average relative error of approximation did not exceed 5%. Similarly, the average absolute error rate was in the range of 10% to 20%, which indicates good accuracy of the forecast with the help of the Fourier series of indicators of the independent variables of the multifactorial linear model of the joint value created in the maritime port sector. All of the determination factors exceeded a value of 95%.

On the basis of the obtained adequate analytical dependencies (Fourier series), we performed a process of approximation, extrapolation, and construction of the upper and lower bounds of the confidence interval for the independent variables of the multifactorial linear model of joint value created in the maritime port sector. The period of bias in the study was two quarters, which did not

exceed 20% of the total length of the dynamic range of each independent variable of the multifactorial linear model of joint value created in the maritime port sector.

## 4. Results

### 4.1. Efficiency of the State Maritime Policy

This article is devoted to the development and substantiation of theoretical/methodological and scientific/practical approaches to the formation of measures of efficiency in the maritime sector. The implementation of effective business models of sustainable development in seaports is aimed at maximizing the creation of the joint value at the macro and micro levels of the economy. The list of factors contributing to the efficiency of the state maritime policy was defined and separated into four groups: group 1 (infrastructural factors): Renewal of port infrastructure; coastal infrastructure of sea stations; ecological and physical safety; and convenience in reaching the port of departure by a cruise ship; group 2 (management factors): The effectiveness of management mechanisms; the level of automation and effective communications technologies; the coordination of various types of transport; and the efficiency of port services; group 3 (marketing factors): Tariff policies (size of tariffs, number of port fees, flexibility of price policy); and competition in the ports; group 4 (service factors): Attractiveness of logistics conditions; the development of international tourism; the development of sea leisure; the development of merchant shipping, shipbuilding, ship repair, and instrument making in the port; and the simplification of port entry procedures.

It has been established that the development the maritime business model at the macroeconomic level in state maritime policy has the following sequence: The desire to participate in global value added chains, which is ensured by the production of high-tech products in the areas of port processing and the provision of high-tech services with high added value; the introduction of complex measures of protectionism and free trade and dependence on the availability of national competitive advantages of the marine port industry for influencing the infrastructure, superstructure, and economic structure of the state maritime policy of Ukraine; and the creation of business models of seaports by basic types (creator, distributor, owner (landlord), broker (broker)) and depending on port type (land feudal lord, tool port, service port). At the same time, the following priorities of building a business model appear: An increase in the amount of taxes paid and the level of employment of the population; an increase in cargo turnover; improvement of the quality of transport services; and an increase in the number of tourists. Consideration of the following complementary assets as the basis for port business model development is suggested: Specialization (reputation, brand, formed clusters, distribution networks, experience and qualifications of experts, expertise, port society, information bases) and generalization (infrastructure, equipment, control facilities (customs processing)), computer systems and automation systems, social networks, ERP-networks, and agreements with state institutions and local self-government bodies). Monitoring of a business model's performance at the macro and micro level should be carried out with information and analytical support to prevent the overall risk of bankruptcy of seaports (with the construction of a risk map). Qualitative and quantitative monitoring takes into account the interests of stakeholders in the maritime industry and is based on the creation of added value in port services as the basis for the shared value of the well-being of all stakeholders.

### 4.2. The Joint Value as a Measure of the Welfare of Stakeholders in the Maritime Industry

The joint value was considered to be a category at both a macroeconomic and microeconomic level, and it was combined with a multivariate regression model performed on the basis of statistical analysis and data processing system Statistics 8.0. The extrapolated data of the independent variables of the multifactor linear model of joint value created in the maritime industry field allowed us to proceed to the extrapolation process of the resulting indicator. The results of the approximation process, extrapolation, and the values of the upper and lower bounds of the confidence interval of the resulting indicator are presented in the Statistica 8.0 tool.

In the course of the calculations, an adequate multifactorial linear model of the joint value created in the marine port sector was obtained. The value of the determination coefficient indicated that the model explained 99.9937% of the total dispersion of its resulting value of the total value of the seaport services. For further investigation into the influence of objective and subjective indicators on the indicator of the joint value of the seaport services, we decided to use a method of expert evaluation, namely the method of analysis of hierarchies.

Using the hierarchy analysis method in the process of evaluating the common value of maritime merchant port services could allow for its decomposition into simple components, prioritizing each and evaluating the level of their interactions with the resulting indicator based on judgments and expert assessments (since they would be substantiated and supported by their experience), which would fill the information gap in the data analysis and significantly reduce the risk of ineffective management decisions regarding the operation of the maritime industry in Ukraine, especially at the macro level.

In the first step, the process of evaluating the joint value of the maritime trading port services in the form of a hierarchy and determining all elements of each level was performed. The first level of the hierarchical model for assessing the total value of the services of the sea trading port consisted of a single element, namely "The purpose is to assess the joint value of the services of the sea trading port".

The elements of the second level were the independent variables of the multifactor linear model of joint value created in the maritime industry, since the relevance of using previous results in the study of the assessment of the joint value of the services of a maritime trading port had already been proven.

## 4.3. Scenarios of Sea Trade Port Industry Development

Based on the results of the fundamental analysis, structural modeling, and synthesis of approaches to business model development, the business model of the seaport was presented as a mechanism for generating a joint value for all stakeholders, considered through optimizing a logically complete set of economic relations within the framework of modeling the development and implementation of the business model. The results of the aggregate economic relations of stakeholders (the state, investors, owners, contractors of ports, the population of the port region, regular workers, invited specialists) within the framework of developing and implementing a business model were the following:

Alternative 1: A scenario of state modernization and corporatization;
Alternative 2: A scenario for managing the seaport as a concession (public–private partnership);
Alternative 3: A scenario for changing the owner of the port subject to attracting private capital; and
Alternative 4: A scenario of inertial development of the port provided there is stagnation in the macroeconomic environment.

The presented alternatives for assessing the total value of the services of the sea trading port were based on the obtained forecast values and the confidence intervals of the independent variables of the multifactor linear model of joint value created in the maritime port sector.

At the third level of the hierarchical model for assessing the joint value of the maritime merchant port services, the above alternatives are presented.

Graphic representation of the hierarchical model for assessing the joint value of the maritime industry is shown in Figure 3.

The next step in working with the hierarchical model for evaluating the joint value of seaport services was to identify local priorities and assess the consistency of judgments.

Using the method of pairwise comparisons, we defined the following indicators: Priorities of the second-level criteria relative to the main goal and priorities of alternatives (scenarios of development) in relation to the second-level criteria.

To this end, we constructed the necessary matrices for pairwise comparisons. For each matrix, we defined the normalized priority vector, the maximal real number, the index, and the relation of the coherence.

The matrix of pairwise comparisons and the estimates of the priorities of the second-level elements in relation to the main goal obtained in its background are given in Table 5.

**Table 5.** Matrix of pairwise comparisons for elements of the second level of the hierarchical model for estimating the joint value of seaport industry services.

| No. | Title of the Elements, Compared at the Second Level of the Hierarchy Model | Element No. | | | | | Local Priorities, $u_i$ |
|---|---|---|---|---|---|---|---|
| | | **1** | **2** | **3** | **4** | **5** | |
| 1 | Profit tax | 1 | 1/3 | 1/5 | 3 | 7 | 0.139 |
| 2 | Net profit | 3 | 1 | 1/3 | 5 | 9 | 0.274 |
| 3 | Staff costs | 5 | 3 | 1 | 5 | 9 | 0.491 |
| 4 | Social fare | 1/3 | 1/5 | 1/5 | 1 | 3 | 0.067 |
| 5 | Depreciation | 1/7 | 1/9 | 1/9 | 1/3 | 1 | 0.030 |

$\lambda_{\max} = 5.309; IU = 0.077; BU = 0.069.$

The components of its own vector of local priorities are calculated by the formulas

$$\overline{u_i} = \sqrt[n]{\prod_{j=1}^{n} a_{ij}}; i = \overline{1,n};, \tag{4}$$

where $a_{ij}$—$i$ is an element of the matrix column of the pairwise criteria comparison and $n$ is the number of criteria, and

$$u_i = \frac{\overline{u_i}}{\sum\limits_{i=1}^{n} \overline{u_i}}; i = \overline{1,n}. \tag{5}$$

Corresponding calculations for the second-level hierarchical model for assessing the value of seaports were as follows:

$$n = 5; \overline{u_1} = \sqrt[5]{1 \cdot \frac{1}{3} \cdot \frac{1}{5} \cdot 3 \cdot 7} = 1.06961;$$

$$\overline{u_2} = \sqrt[5]{3 \cdot 1 \cdot \frac{1}{3} \cdot 5 \cdot 9} = 2.141127;$$

$$\overline{u_3} = \sqrt[5]{5 \cdot 3 \cdot 1 \cdot 5 \cdot 9} = 3.68011;$$

$$\overline{u_4} = \sqrt[5]{\frac{1}{3} \cdot \frac{1}{5} \cdot \frac{1}{5} \cdot 1 \cdot 3} = 0.525306,$$

$$\overline{u_5} = \sqrt[5]{\frac{1}{7} \cdot \frac{1}{9} \cdot \frac{1}{9} \cdot \frac{1}{3} \cdot 1} = 0.22587,$$

$$\sum_{i=1}^{5} \overline{u_i} = 1.06961 + 2.141127 + 3.68011 + 0.525306 + 0.22587 = 7.642023,$$

$$u_1 = \frac{1.06961}{7.642023} = 0.139,$$

$$u_2 = \frac{2.141127}{7.642023} = 0.274,$$

$$u_3 = \frac{3.68011}{7.642023} = 0.491,$$

$$u_4 = \frac{0.525306}{7.642023} = 0.067,$$

$$u_5 = \frac{0.22587}{7.642023} = 0.030.$$

The maximal proper value of the inverse-symmetric matrix of pairwise comparisons was determined by the following formula:

$$\lambda_{\max} \approx \sum_{j=1}^{n} u_j \left( \sum_{i=1}^{n} a_{ij} \right); i = \overline{1, n}. \tag{6}$$

For elements of the second level of the hierarchical model of estimating the joint value of seaport industry services, we defined the following:

$$\sum_{i=1}^{5} a_{i1} = 1 + \frac{1}{3} + \frac{1}{5} + 3 + 7 = 11.53,$$

$$\sum_{i=1}^{5} a_{i2} = 3 + 1 + \frac{1}{3} + 5 + 9 = 18.33,$$

$$\sum_{i=1}^{5} a_{i3} = 5 + 3 + 1 + 5 + 9 = 23.00,$$

$$\sum_{i=1}^{5} a_{i4} = \frac{1}{3} + \frac{1}{5} + \frac{1}{5} + 1 + 3 = 4.73,$$

$$\sum_{i=1}^{5} a_{i5} = \frac{1}{7} + \frac{1}{9} + \frac{1}{9} + \frac{1}{3} + 1 = 1.70,$$

$$\lambda_{\max} \approx 5.309.$$

The coherence (homogeneity) of the matrices, which reflects the imitation of an expert's logic in expressing his own judgments, is quite important in the process of constructing matrices of pairwise comparisons.

As an indicator of the degree of consistency of the elements of the matrix of pairwise comparisons, a homogeneity index (index of coherence) was used. It was calculated by the formula

$$IO = I\text{У} = \frac{\lambda_{\max} - n}{n - 1}. \tag{7}$$

To assess the acceptability of the degree of consistency of the matrix elements, the ratio of homogeneity (consistency), which was calculated by the following formula, was used:

$$BO = B\text{У} = \frac{IO}{M(IO)}, \tag{8}$$

where $M(IO)$ is the average value of the homogeneity index of a randomly composed matrix of pairwise comparisons, based on experimental data (for $n = 5$ the table meaning equals $M(IO) = 1, 12$).

It was acceptable for further use of the obtained matrices of pairwise comparisons to be the value of the homogeneity relation (consistency), which was less than or equal to 0.1, (that is, $BO \leq 0, 10$). The excess of the index of homogeneity (coherence) of the value of 0,1 indicated a significant violation of the logic of judgments, which the expert assumed when filling matrices of pairwise comparisons, so the expert should be asked to revise the data to improve this indicator. The result of the matrix of pairwise comparison of elements of the second level $BO$ did not exceed a value of 0.1, which testified to the loyalty of the judgment logic of the research expert and the possibility of its use in subsequent calculations of the hierarchical model for evaluating the joint value of the services of the sea trading port.

In the next stage, we filled in matrices of pairwise comparisons for the third-level elements (alternatives) of the hierarchical model for assessing the total value of the services of sea merchant ports according to all of the criteria of the second level, and we determined their local priorities. The matrices of pairwise comparisons of elements of the third level according to all of the criteria of the second level of the hierarchical model for assessing the total value of services in the maritime industry are presented in Tables 6–10.

**Table 6.** Matrix of pairwise comparisons for the elements of the third level by the criteria "profit tax".

| No. | Elements to Be Compared in the Third Level of the Hierarchy Model | Element No. | | | | Local Priorities, $V_{i1}$ |
|---|---|---|---|---|---|---|
| | | 1 | 2 | 3 | 4 | |
| 1 | Alternative 1 | 1 | 3 | 5 | 7 | 0.565 |
| 2 | Alternative 2 | 1/3 | 1 | 3 | 5 | 0.262 |
| 3 | Alternative 3 | 1/5 | 1/3 | 1 | 3 | 0.118 |
| 4 | Alternative 4 | 1/7 | 1/5 | 1/3 | 1 | 0.055 |

$\lambda_{max} = 4.119$; $IY = 0.040$; $BY = 0.044$.

**Table 7.** Matrix of pairwise comparisons for the elements of the third level by the criteria "net profit".

| No. | Elements to Be Compared in the Third Level of the Hierarchy Model | Element No. | | | | Local Priorities, $V_{i2}$ |
|---|---|---|---|---|---|---|
| | | 1 | 2 | 3 | 4 | |
| 1 | Alternative 1 | 1 | 3 | 7 | 9 | 0.594 |
| 2 | Alternative 2 | 1/3 | 1 | 5 | 3 | 0.257 |
| 3 | Alternative 3 | 1/7 | 1/5 | 1 | 3 | 0.094 |
| 4 | Alternative 4 | 1/9 | 1/3 | 1/3 | 1 | 0.056 |

$\lambda_{max} = 4.254$, $IY = 0.085$, $BY = 0.094$.

**Table 8.** Matrix of pairwise comparisons for the elements of the third level by the criteria "Staff costs".

| No. | Elements to Be Compared in the Third Level of the Hierarchy Model | Element No. | | | | Local Priorities, $V_{i3}$ |
|---|---|---|---|---|---|---|
| | | 1 | 2 | 3 | 4 | |
| 1 | Alternative 1 | 1 | 3 | 5 | 5 | 0.549 |
| 2 | Alternative 2 | 1/3 | 1 | 3 | 3 | 0.248 |
| 3 | Alternative 3 | 1/5 | 1/3 | 1 | 3 | 0.129 |
| 4 | Alternative 4 | 1/5 | 1/3 | 1/3 | 1 | 0.074 |

$\lambda_{max} = 4.202$; $IY = 0.067$; $BY = 0.075$.

**Table 9.** Matrix of pairwise comparisons for the elements of the third level by the criteria "social fare".

| No. | Elements to Be Compared in the Third Level of the Hierarchy Model | Element No. | | | | Local Priorities, $V_{i4}$ |
|---|---|---|---|---|---|---|
| | | 1 | 2 | 3 | 4 | |
| 1 | Alternative 1 | 1 | 5 | 9 | 9 | 0.680 |
| 2 | Alternative 2 | 1/5 | 1 | 3 | 3 | 0.177 |
| 3 | Alternative 3 | 1/9 | 1/3 | 1 | 3 | 0.091 |
| 4 | Alternative 4 | 1/9 | 1/3 | 1/3 | 1 | 0.052 |

$\lambda_{max} = 4.191$; $IY = 0.064$; $BY = 0.071$.

On the basis of the obtained results, we used the synthesis principle to determine the global priorities of the third-level elements. They were defined as the sum of applications of the local priorities of each element at the third level to the global priorities of the second-level elements of the hierarchical model for evaluating the joint value of the services of the maritime industry:

$$W_i = V_{ij} \cdot u_i, \tag{9}$$

where $V_{ij}$ is the local priority of the *i* element of the third level in relation to the *j* element of the second level.

**Table 10.** Matrix of pairwise comparisons for the elements of the third level by the criteria "depreciation".

| No. | Elements to Be Compared in the Third Level of the Hierarchy Model | Element No. | | | | Local Priorities, $V_{i5}$ |
| --- | --- | --- | --- | --- | --- | --- |
| | | 1 | 2 | 3 | 4 | |
| 1 | Alternative 1 | 1 | 5 | 3 | 9 | 0.599 |
| 2 | Alternative 2 | 1/5 | 1 | 3 | 3 | 0.211 |
| 3 | Alternative 3 | 1/3 | 1/3 | 1 | 3 | 0.134 |
| 4 | Alternative 4 | 1/9 | 1/3 | 1/3 | 1 | 0.056 |

$\lambda_{\max} = 4.265$; $IУ = 0.088$; $BУ = 0.098$.

For the submitted alternatives at the third level of the hierarchical model for assessing the joint value of the services of the maritime industry of Ukraine:

$$W_1 = 0.139 \cdot 0.565 + 0.274 \cdot 0.594 + 0.491 \cdot 0.549 + 0.067 \cdot 0.680 + \\ + 0.030 \cdot 0.599 = 0.574;$$

$$W_2 = 0.139 \cdot 0.262 + 0.274 \cdot 0.257 + 0.491 \cdot 0.248 + 0.067 \cdot 0.177 + \\ + 0.030 \cdot 0.211 = 0.246;$$

$$W_3 = 0.139 \cdot 0.118 + 0.274 \cdot 0.084 + 0.491 \cdot 0.129 + 0.067 \cdot 0.091 + \\ + 0.030 \cdot 0.134 = 0.115.$$

$$W_4 = 0.139 \cdot 0.055 + 0.274 \cdot 0.056 + 0.491 \cdot 0.074 + 0.067 \cdot 0.052 + \\ + 0.030 \cdot 0.056 = 0.064.$$

The complex combination of results from the multifactor linear model of joint value created in the maritime port sector and the hierarchical model for assessing the joint value of the maritime merchant port services allowed us to obtain a calculated confirmation of the best further alternative for developing the port industry in Ukraine: Namely, a scenario of state modernization and corporatization.

## 5. Conclusions

The creation of a joint value as a measure of the welfare of stakeholders in the maritime industry was measured by using a statistical and mathematical regression model based on statistics. The elements of the effective indicator were income tax on general activities, net profit, payment of wages, deductions for social events, and amortization. This model allowed us to take into account the causal relationships of changes among independent variables outlined by the study and the result indicator, and acted as a convenient tool in the process of forming operational plans of activity and risk management of individual seaports and the port industry as a whole.

The joint value was considered to be a category at both a macroeconomic and microeconomic level, because it combined endogenous and exogenous factors of influence. The complex combination of results from the multifactorial linear model of joint value created in the maritime industry led to the conclusion that the best alternative to the development of the port industry in Ukraine is the scenario of state modernization and corporatization.

The added value of the study for practice is the development of a maritime industry effect measurement relevant to each of its stakeholders. The contribution to the literature concerns a new interpretation of the industry effect implemented in the joint value.

**Author Contributions:** Conceptualization, I.N.; methodology, N.R.; software, N.R.; validation, I.N.; formal analysis, I.N.; investigation, I.N.; resources, I.N.; data curation, I.N.; writing—original draft preparation, I.N.; writing—review and editing, S.M.; visualization, N.R.; supervision, N.R.; project administration, I.N.; funding acquisition, I.N.

**Funding:** This research received no external funding.

**Conflicts of Interest:** The authors declare no conflicts of interest.

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
