# Peer review of "Joint Value as a Measure of Sea Trade Port Stakeholder Effect"

_socsci, doi:10.3390/socsci8040120_

Round 1
Reviewer 1 Report
The paper is not well written and in several instances it is quite difficult to understand what the authors had in mind. After reading the introduction, the reader can only suppose the core of the paper.
The article is devoted to the development and substantiation of theoretical-methodological and scientific-practical approaches to the formation of the measures of efficiency in the maritime sector. Anyway, neither the theoretical model nor methodological approach are adequately discussed; therefore, I have many reservations about its contribution to the literature and its technical accuracy.
In my opinion a good bit of work needs to be done to make clearly defined the goals and the empirical methodology.
Author Response
Dear Professor,
thank you very much for your review. We tried to do our best to correct the attached article.
Response to Reviewer 1 Comments
The paper is not well written and in several instances it is quite difficult to understand what the authors had in mind. After reading the introduction, the reader can only suppose the core of the paper.
Response 1: The introduction content was added to clarify the results of the research.Lines 30 – 60.
The article is devoted to the development and substantiation of theoretical-methodological and scientific-practical approaches to the formation of the measures of efficiency in the maritime sector. Anyway, neither the theoretical model nor methodological approach are adequately discussed; therefore, I have many reservations about its contribution to the literature and its technical accuracy.
Response 2: The methodological approach was developed and new structure of the article introduced. The literature review was rewritten (lines 66 - 81).
In my opinion a good bit of work needs to be done to make clearly defined the goals and the empirical methodology.
Response 3: The goals and tasks were formulated specifically to express the logic of the article.

Reviewer 2 Report
As a general comment, this paper needs a major revision work before it can be published.
Abstract
The aim of the study should be clearly presented in the paper.
“How to maximize the creation of a joint value in the process of implementation of business-models of sustainable development of sea ports of Ukraine.” Is this the aim of the paper?
Structure of the paper
This paper needs a in-depth reorganization of its structure.
Why is the results section included before the materials and method section?
Specifically, the paper should include the following sections: 1) introduction; 2) literature review; 3) research model and hypotheses; 4) methodology; 5) sample description; 6) results; 7) conclusions. Particularly, authors should presents reasons that justify their study and the main goal in the introduction section, emphasizing the added value/contribution of their study to literature and practice. These issues have not been discussed by authors. As I understand, this paper is focused on sea trade market but there is no reference to this industry in the introduction section. Why such industry has relevance with respect to the research model the authors want to investigate and empirically test? The literature review section is necessary to develop arguments on which the hypotheses are based. The hypotheses should be developed in section 3) research model and hypotheses using main references presented in section 2).
Authors refer to the following theories: theories of stakeholders, labor cost and marginal utility….However, they do not illustrate them. How, the research model is developed from such theories?
What is the methodology adopted by the authors? It seems that authors use AHP and statistical inference (linear regression) as data analysis technique. Why such choice? Are these techniques part of the methodology used to test hypotheses? There are no justifications that explain the reasons the authors adopted these methods. Why do authors use different type of analytical dependence? The analytical dependence should be the result of theoretical reasoning rather than an issue linked to the attempt to find statistically significant relationships between variables.
Section 3 (research model and hypotheses) should present the main variables of the model. Why such variables are important? Are they proposed taking into account an in-depth literature survey?
Line 47: the efficiency concept has not been presented before. Why is it important to the study? If it is a critical concept, i.e. a leading concept to carry on the overall study, it should presented in the introduction section and discussed with greater details in the research model and hypotheses section (section 3).
Line 357: “… dissertation study…”: is this paper part of a PhD dissertation?
Author Response
Dear Professor,
thank you very much for your deep review. We corrected the attached article according your comments.
Response to Reviewer 2 Comments
Abstract
The aim of the study should be clearly presented in the paper.
“How to maximize the creation of a joint value in the process of implementation of business-models of sustainable development of sea ports of Ukraine.” Is this the aim of the paper?
Response 1:
Structure of the paper
This paper needs a in-depth reorganization of its structure.
Why is the results section included before the materials and method section?
Specifically, the paper should include the following sections: 1) introduction; 2) literature review; 3) research model and hypotheses; 4) methodology; 5) sample description; 6) results; 7) conclusions.
Response 2: The new structure of the paper is developed in the article.
Particularly, authors should presents reasons that justify their study and the main goal in the introduction section, emphasizing the added value/contribution of their study to literature and practice. These issues have not been discussed by authors. As I understand, this paper is focused on sea trade market but there is no reference to this industry in the introduction section. Why such industry has relevance with respect to the research model the authors want to investigate and empirically test? The literature review section is necessary to develop arguments on which the hypotheses are based. The hypotheses should be developed in section 3) research model and hypotheses using main references presented in section 2).
Response 3: The research hypothesis is presented accordingly – lines 111-118.
Authors refer to the following theories: theories of stakeholders, labor cost and marginal utility….However, they do not illustrate them. How, the research model is developed from such theories?
Response 4: The background theories were shortened and the aim become specific while the reseach model is described in the methodology section.
What is the methodology adopted by the authors? It seems that authors use AHP and statistical inference (linear regression) as data analysis technique. Why such choice? Are these techniques part of the methodology used to test hypotheses? There are no justifications that explain the reasons the authors adopted these methods. Why do authors use different type of analytical dependence? The analytical dependence should be the result of theoretical reasoning rather than an issue linked to the attempt to find statistically significant relationships between variables.
Section 3 (research model and hypotheses) should present the main variables of the model. Why such variables are important? Are they proposed taking into account an in-depth literature survey?
Response 5:The variables choice is explained in lines 111 – 118.
Line 47: the efficiency concept has not been presented before. Why is it important to the study? If it is a critical concept, i.e. a leading concept to carry on the overall study, it should presented in the introduction section and discussed with greater details in the research model and hypotheses section (section 3).
Response 6: The explanation is presented in the introduction section.
Line 357: “… dissertation study…”: is this paper part of a PhD dissertation?
Response 7: The text was corrected.

Reviewer 3 Report
The main purpose of this research is to develop a theoretical and practical approach for investigating efficiency measures in the maritime sector. It aims to maximize the creation of joint value at the macro and micro levels of the economy. In this framework, the factors contributing to the efficiency of the state maritime policy were defined and separated into four groups. This is an interesting research work and the topic is important for the readers of the journal.
However, some concerns arise regarding the manuscript. The research questions as well as the original contribution of the work, comparing to other previous works are not well clarified. The authors should focus on the theoretical background and analysis of the approach, explaining the steps of the research and the model parameters. A better explanation of the results is also needed. In addition, a further analysis of related research work is needed.
From a methodological point of view, I would expect a justification regarding the applied analysis. A sensitivity analysis of the results would provide a more solid basis for the approach. Further, a considerable reorganization of the manuscript is needed with a more logical flow of the sections.
Additional comments and recommendations for the improvement of the manuscript:
General notes
Ø The English language should be improved throughout the manuscript.
Ø In-text citations include the reference number enclosed in square brackets or the author names in running text and the date of publication inside parentheses.
Ø All the equation symbols must be defined within the manuscript.
Ø All the equations should be numbered properly.
Ø Please check the text format. Fond size and type should be corrected.
Ø Many references are not mentioned in the text.
Abstract
[Lines 12-21] The groups of factors which contribute to the efficiency of the state maritime policy and to the development of the maritime industry shouldn’t be presented in details in this section.
Introduction
General note: This section is too short and the purpose of the current research is not adequately presented. A more critical literature review is needed and the research gap should be clarified.
[Lines 34-35] “The attention of such scientists as Chesbrough (2007), Schweizer (2005), Debelak (2006), Sliwotsky (2006) is directed…” instead of “The attention of such scientists as Chesbrough (Chesbrough 2007), Schweizer (Schweizer 2005), Debelak (Debelak 2006), Sliwotsky (Sliwotsky 2006) is directed…”
[Line 36] “One of the still unsolved tasks…” instead of “One of the still unsolved task…”
[Lines 37, 40 and 44] Same comment as in lines 34-35.
Results
General notes:
(1) The “results” should be presented after the experimental analysis. The title of this section could be changed.
(2) It is not necessary to present all the calculations in your manuscript.
(3) It would be preferable to use a decimal point instead of a decimal comma. E.g.
(4) The same number of decimal digits should be used in lines 178-182
[Fig. 1] “Profit tax (0.139)” instead of “Profit tax (0,139)”. The numbers should be explained.
[Line 87] “…a multivariate regression model performed on the basis of statistical analysis…” instead of “…a multivariate regression model performed on the basis of statistical statistical analysis…”.
[Line 104] “…the confidence interval of the resulting indicator are presented…” instead of “…the confidence interval of the resulting indicator presented…”
[Table 1] Units are missing.
[Line 224] “…are presented in Tables 2-6…” instead of “…are presented in Table. 2-6…”
Materials and Methods
[Lines 267-268] The equation’s numbering should be corrected.
[Table 7] “x1” instead of “x1” etc. Units are missing.
[Lines 289-290] Please place “Fig. 2” properly.
[Fig. 2] The figures are not clear. Units are missing. Legend is missing.
[Lines 303-304] “…in the Statistica 8.0 package are obtained. table 8 is presented…”. This sentence should be corrected.
[Lines 310-311, 324-325] The equations’ numbering should be corrected.
Conclusions
General note: In this section, the original contribution of the research has to be presented by focusing on the research results based on the research questions.
[Title] “Conclusions” instead of “Conclusions.”.
[Lines 364-365] “…is considered both as a category and a macro. and the micro level…”. This sentence needs to be revised.
References
Many references are not mentioned in the text:
Abuzyarova, M. 2015, Alderton, P. 2008, Batocchio, A.; Minatogawa, V. 2017, Bossidi, L., Charan, R. 2007, Buyanov, D. 2014, Kim, W. Chan, and Renee A. Mauborgne. 2014, Dolzhenkova, E. Kazakova, M. 2015, Osterwalder, A., Pigneur, Y. 2010, Pastor-Agustín, G., Marisa Ramírez-Alesón, M., Espitia-Escuer, M. 2011, Rothaermel, F., Hill Charles, W. 2005, Rothaermel, F.T. 2001, Slywotzky, A. J. 2006, Tripsas, M. 1997, Pek-Hooi, S., Jiang, Y. 2010, Vleugels, R.L.M. 1969, Zott, C., Amit, R., Massa, L. 2011.
Author Response
Dear Professor, thank you very much for your your expert review. We tried to do our best to correct the article. Your recommendations were very usefull.
Response to Reviewer 3 Comments
The main purpose of this research is to develop a theoretical and practical approach for investigating efficiency measures in the maritime sector. It aims to maximize the creation of joint value at the macro and micro levels of the economy. In this framework, the factors contributing to the efficiency of the state maritime policy were defined and separated into four groups. This is an interesting research work and the topic is important for the readers of the journal.
However, some concerns arise regarding the manuscript. The research questions as well as the original contribution of the work, comparing to other previous works are not well clarified. The authors should focus on the theoretical background and analysis of the approach, explaining the steps of the research and the model parameters. A better explanation of the results is also needed. In addition, a further analysis of related research work is needed.
Response 1: The analysis of he related research works was made additionally in the literature revie and introduction.
From a methodological point of view, I would expect a justification regarding the applied analysis. A sensitivity analysis of the results would provide a more solid basis for the approach. Further, a considerable reorganization of the manuscript is needed with a more logical flow of the sections.
Response 2: The reorganization of the article was made.
Additional comments and recommendations for the improvement of the manuscript:
General notes
Ø In-text citations include the reference number enclosed in square brackets or the author names in running text and the date of publication inside parentheses.
Ø All the equation symbols must be defined within the manuscript.
Ø All the equations should be numbered properly.
Ø Please check the text format. Fond size and type should be corrected.
Ø Many references are not mentioned in the text.
Response 3: All the recommendations were taken into account.
Results
General notes:
(1) The “results” should be presented after the experimental analysis. The title of this section could be changed.
(2) It is not necessary to present all the calculations in your manuscript.
(3) It would be preferable to use a decimal point instead of a decimal comma. E.g.
(4) The same number of decimal digits should be used in lines 178-182
[Fig. 1] “Profit tax (0.139)” instead of “Profit tax (0,139)”. The numbers should be explained.
[Line 87] “…a multivariate regression model performed on the basis of statistical analysis…” instead of “…a multivariate regression model performed on the basis of statistical statistical analysis…”.
[Line 104] “…the confidence interval of the resulting indicator are presented…” instead of “…the confidence interval of the resulting indicator presented…”
[Table 1] Units are missing.
[Line 224] “…are presented in Tables 2-6…” instead of “…are presented in Table. 2-6…”
Response 4: The structure of the paper was changed. All the text was corrected according mentioned lines mistakes.
Materials and Methods
[Lines 267-268] The equation’s numbering should be corrected.
[Table 7] “x1” instead of “x1” etc. Units are missing.
[Lines 289-290] Please place “Fig. 2” properly.
[Fig. 2] The figures are not clear. Units are missing. Legend is missing.
[Lines 303-304] “…in the Statistica 8.0 package are obtained. table 8 is presented…”. This sentence should be corrected.
[Lines 310-311, 324-325] The equations’ numbering should be corrected.
Response 5: All the lines were corrected.
Conclusions
General note: In this section, the original contribution of the research has to be presented by focusing on the research results based on the research questions.
[Title] “Conclusions” instead of “Conclusions.”.
[Lines 364-365] “…is considered both as a category and a macro. and the micro level…”. This sentence needs to be revised.
Response 6:The original contribution was described – lines 411 -425. All the corrections were made.
References
Many references are not mentioned in the text:
Abuzyarova, M. 2015, Alderton, P. 2008, Batocchio, A.; Minatogawa, V. 2017, Bossidi, L., Charan, R. 2007, Buyanov, D. 2014, Kim, W. Chan, and Renee A. Mauborgne. 2014, Dolzhenkova, E. Kazakova, M. 2015, Osterwalder, A., Pigneur, Y. 2010, Pastor-Agustín, G., Marisa Ramírez-Alesón, M., Espitia-Escuer, M. 2011, Rothaermel, F., Hill Charles, W. 2005, Rothaermel, F.T. 2001, Slywotzky, A. J. 2006, Tripsas, M. 1997, Pek-Hooi, S., Jiang, Y. 2010, Vleugels, R.L.M. 1969, Zott, C., Amit, R., Massa, L. 2011.
Response 7: The corrections were made.

Round 2
Reviewer 1 Report
The paper is greatly improved.
Author Response
Dear reviewer, thank you very much for your review. I did my best to improve the paper and grayefull for your valuable recommendations.
Reviewer 2 Report
Authors have made an effort to improve their paper. However, there are still some weaknesses that need to be addressed.
Line 108: “According to the research hypothesis these elements are relevant to the effect obtained by the…”. Where is the research hypothesis? Authors did not presented any hypothesis before. They should state their main hypothesis at the end of the literature section. Additionally the hypothesis should clearly present the relationship between two or more variables, one of them associated to a performance measure (i.e., efficiency, joint value, etc.), the other variables associated to factors affecting performance. Finally, this hypothesis should be linked to the main word used by authors in the paper title, i.e. joint value and stakeholder effect.
Table 1: how did authors get a measurement of the joint value concept used in the regression analysis? There is no indication about that in the paper before the presentation of this table.
It is unclear if most of the joint value effect is linked to efficiency. Please, discuss such issue. It seems that sometimes authors privilege efficiency.
I suggest that authors include a flow chart that illustrates the steps of the overall methodology used in the paper to generate results.
Author Response
Dear Sir/ Madam,
I am very gratefull for your patience and deep analysis of my work. beacus of your review I rework the article deeply. Hope this attempt will reach the goal.
Deep regards,
Reviewer 3 Report
In the revised edition, the manuscript has been significantly improved and covered my main comments.
Author Response
Dear reviwer, I am very gratefull. Because of your recommendations my article became more qualitative.
Round 3
Reviewer 2 Report
This paper still needs some work to make it ready for publication.
- please, revise English style and grammar (including the Abstract).
- as I emphasized in my previous review, the relationship between the efficiency concept and the joint value effect should be discussed providing a more in-depth argumentation.
- variables used in the analysis (see lines 115 - 119) have not been included in the previous section 2. The authors should clearly present the model they want to implement in the empirical analysis to readers. Thus, these variables should emerge from the discussion in section 2.
Author Response
Dear Reviewer,
thank you very much for your patience and in-depth analysis of our paper. Attached we send the corrected text with the responses to all comments.
Deep regards
